# The Modular Architecture of Metallothioneins Facilitates Domain Rearrangements and Contributes to Their Evolvability in Metal-Accumulating Mollusks

**DOI:** 10.3390/ijms232415824

**Published:** 2022-12-13

**Authors:** Sara Calatayud, Mario Garcia-Risco, Veronika Pedrini-Martha, Michael Niederwanger, Reinhard Dallinger, Òscar Palacios, Mercè Capdevila, Ricard Albalat

**Affiliations:** 1Departament de Genètica, Microbiologia i Estadística, Facultat de Biologia, Universitat de Barcelona, E-08028 Barcelona, Spain; 2Departament de Química, Facultat de Ciències, Universitat Autònoma de Barcelona, E-08193 Cerdanyola del Vallès, Spain; 3Center for Molecular Biosciences Innsbruck (CMBI), Department of Zoology, University of Innsbruck, A-6020 Innsbruck, Austria; 4Institut de Recerca de la Biodiversitat (IRBio), Universitat de Barcelona, E-08028 Barcelona, Spain

**Keywords:** protein modularity, domain repeat proteins, de novo domains, sORFs and lncRNAs, Cephalopoda and Caudofoveata MTs

## Abstract

Protein domains are independent structural and functional modules that can rearrange to create new proteins. While the evolution of multidomain proteins through the shuffling of different preexisting domains has been well documented, the evolution of domain repeat proteins and the origin of new domains are less understood. Metallothioneins (MTs) provide a good case study considering that they consist of metal-binding domain repeats, some of them with a likely de novo origin. In mollusks, for instance, most MTs are bidomain proteins that arose by lineage-specific rearrangements between six putative domains: α, β1, β2, β3, γ and δ. Some domains have been characterized in bivalves and gastropods, but nothing is known about the MTs and their domains of other Mollusca classes. To fill this gap, we investigated the metal-binding features of NpoMT1 of *Nautilus pompilius* (Cephalopoda class) and FcaMT1 of *Falcidens caudatus* (Caudofoveata class). Interestingly, whereas NpoMT1 consists of α and β1 domains and has a prototypical Cd^2+^ preference, FcaMT1 has a singular preference for Zn^2+^ ions and a distinct domain composition, including a new Caudofoveata-specific δ domain. Overall, our results suggest that the modular architecture of MTs has contributed to MT evolution during mollusk diversification, and exemplify how modularity increases MT evolvability.

## 1. Introduction

A protein domain is a well-defined region of a protein that constitutes a stable, independently folding, and compact structural unit that might perform a specific function [1,2,3]. Protein domains may occur alone, but are more frequently found in combination with other domains in multidomain proteins [4,5,6]. Novel multidomain proteins might arise by rearranging preexisting domains or by merging de novo emerged domains with old domains. While domain rearrangements have been analyzed extensively ([4,6,7,8,9,10,11]; reviewed in [12,13]), the rise and evolution of new domains are still poorly understood [14,15,16], and examples in which the new domains have been functionally characterized are very scarce [17]. In this scenario, metallothioneins (MTs), a heterogeneous family of proteins made of metal-binding domains, are becoming a valuable case study to infer events of the de novo emergence of domains and to understand the impact of a modular architecture on the evolution of novel proteins.

MTs have classically been defined as low molecular weight (<100 amino acids) and cysteine-rich (≈15–30%) proteins that, because they bind transition metal ions and lack enzymatic functions, have been related with metal homeostasis and detoxification in living beings ([18,19]; reviewed in [20]). Their cysteine residues (Cys, C) are arranged in CC, CxC or CCC motifs, whose number and distribution define the structural domains that form metal–thiolate clusters. In mollusks, for instance, MT domains have been classified into six types: α domains (with 11/12 Cys), β1, β2 and β3 domains (with 9 Cys, each one with different Cys motif arrangements), γ domains (with 10 Cys) and δ domains (with 14 Cys) [21]. Sequence analyses have led to suggest that most mollusk MTs would be bidomain proteins that combine an invariable carboxy-terminal β1 domain with one of the other five domains (α, β2, β3, γ or δ) in a lineage-specific manner: an α domain in some Bivalvia, Cephalopoda, Polyplacophora and Solenogastres (also known as Neomeniamorpha) MTs, a β2 domain in several Bivalvia, Monoplacophora, Scaphopoda and Solenogastres MTs, a β3 domain in all Gastropoda MTs except in Patellogastropoda MTs that contain a γ domain, and a δ domain only found in Caudofoveata, (also known as Chaetodermomorpha) MTs [21,22,23]. 

Mollusk β and γ domains have been functionally characterized, and their metal preferences as well as their binding capacities have been determined [21,24]. In contrast, functional information on the α and δ domains is scarce or totally absent. To fill this gap, we have now analysed the MTs of two singular mollusk species, the “living fossil” *Nautilus pompilius* of the Cephalopoda class, and *Falcidens caudatus*, a Caudofoveata species of the Aculifera group that includes worm-like mollusk species that have no shell. Based on their sequences, the MTs of both species were predicted to be bidomain MTs, with α and β1 domains for the *N. pompilius* MT (i.e., αβ1-NpoMT1), and δ and β1 domains for the *F. caudatus* MT (i.e., δβ1-FcaMT1) [21]. In this study, we determined the metal-binding abilities of both αβ1- and δβ1-MTs characterizing by means of spectrometric (ESI-MS) and spectroscopic (ICP-AES) techniques the metal–protein complexes when these MTs are heterologously expressed in *E. coli*. We also analysed the features of α and δ domains when expressed alone, demonstrating their functional and structural independence to form metal–protein complexes. Our analyses highlight the relevance of the modular organization in MT evolvability by showing how functionally autonomous domains might arise and recombine into new MTs that have been adaptively selected according to the metal requirements and bioavailability in different mollusk lineages.

## 2. Results

### 2.1. Characterization of Cephalopoda and Caudofoveata MTs: NpoMT1 and FcaMT1

NpoMT1 from *N. pompilius* and FcaMT1 of *F. caudatus* were selected as representative Cephalopoda and Caudofoveata MTs, respectively. NpoMT1 was a 72 amino acid protein with 21 Cys (29%) organized in two putative domains: an amino-terminal α domain with 12 Cys arranged in a pattern of [CxC]x_5_[CxC]x_3_Cx_4_[CxC]x_3_[CxC]x_3_[CxC]x_2_C motifs, linked by three residues to a carboxyl-terminal β1 domain with its 9 Cys arranged in a pattern of [CxC]x_3_[CxC]x_3_Cx_5_[CxC]x_3_[CxC] motifs (Figure 1). The patterns of Cys motifs in the α and β1 domains of *Nautilus* NpoMT1 were more similar to those in other mollusk α and β1 domains than those of other cephalopod species such as *Sepia* and *Octopus,* suggesting that they represented the ancestral forms [21]. FcaMT1 was a 74 amino acid protein with 23 Cys (31%) organized in two predicted domains: an amino-terminal δ domain with 14 Cys arranged in a pattern of [CxC]x_3_[CxCCC]x_4_Cx_4_[CxC]x_3_CCx_4_[CxCC] motifs, linked by three residues to a carboxyl-terminal archetypal β1 domain (Figure 1). Importantly, the δ domain has so far only been found in Caudofoveata MTs, suggesting a lineage-specific innovation [21].

In order to characterize the biochemical properties and metal-binding features of NpoMT1 and FcaMT1, we took advantage of a well-established procedure for MT analysis, which determines the metal preference and binding capacity of any MT (or MT domain) based on the study of metalated species in recombinantly produced metal–protein complexes [20]. Thus, both proteins were expressed as GST-fusion proteins in *E. coli* BL21 cultures supplemented with ZnCl_2_, CdCl_2_ or CuSO_4_. Metal–protein complexes were purified from *E. coli* total protein extracts by a GST-affinity system, followed by the cleavage with thrombin of the GST tag, and FPLC chromatography. The FPLC fractions containing the metal-MT complexes were characterized by ESI-MS analyses. The experimental masses corresponding to apo-NpoMT1 and apo-FcaMT1 (7471.8 Da and 7593.6 Da, respectively) after demetallation by acidification of the NpoMT1 and FcaMT1 productions were consistent with the theoretical masses (7472.52 Da and 7594.79 Da, respectively) (Figure 2A,C), verifying the identity of the recombinantly expressed MTs. 

The ICP-AES and ESI-MS analyses of the NpoMT1 produced in cadmium (Cd)-enriched culture medium yielded a unique metalated species of NpoMT1 coordinating seven Cd^2+^ ions (Cd_7_-NpoMT1, Figure 3A). This result indicated an efficient and thermodynamically favored binding of NpoMT1 with Cd^2+^ ions. This fact was corroborated by the reluctance of some metal clusters to exchange their metals by protons in acidic conditions (see Cd_4_-NpoMT1 in Figure 2A). In contrast, when NpoMT1 was produced in zinc (Zn) surplus conditions, it rendered a mixture of Zn_6_- and Zn_7_-NpoMT1 as major species (Figure 3B), suggesting that although NpoMT1 binds both divalent metal ions, Cd^2+^ and Zn^2+^, it had a higher preference for Cd^2+^ than for Zn^2+^ ions. These results were additionally supported by the presence of small but perceptible amounts of glycosylated MT species in the productions under Zn^2+^ surplus conditions (peak denoted by an asterisk in Figure 3B), which is characteristic of partly structured Cd-thioneins when produced in absence of their preferred metal [27]. Finally, the productions of NpoMT1 in copper (Cu) surplus conditions also sustained its Cd-specificity. Cd-thioneins typically yield a great variety of homometallic Cu-MT complexes when expressed in Cu-enriched medium, whereas Zn-thioneins produce heterometallic Zn/Cu-MT complexes [24,28]. Our results clearly showed many homometallic Cu-NpoMT1 complexes, confirming its Cd-thionein character (Figure 3C). The Cd preference of NpoMT1 agrees with the association between MT expression and the elevated levels of Cd (but not Zn) reported for *Octopus vulgaris,* another cephalopod species [29]. Based on all this evidence, we classified NpoMT1 as a genuine Cd-thionein.

Regarding FcaMT1, the analysis of metal-FcaMT1 complexes revealed some interesting differences in comparison with NpoMT1. A unique metal-MT species (Zn_8_- and Cd_8_-FcaMT) was observed when FcaMT1 was synthesized in Zn^2+^- or Cd^2+^-enriched medium (Figure 3D,E). This result contrasted with the mixture of Zn^2+^-MT species observed for NpoMT1, and suggested that FcaMT1 was able to bind both divalent metal ions with a similar efficiency. To clarify whether FcaMT1 had a more Zn-thionein or Cd-thionein character, we took advantage of additional information provided by the results of the glycosylation analysis and the productions under Cu surplus conditions. The observation of heterometallic Zn,Cu-FcaMT1 complexes when expressed in Cu-enriched medium (Figure 3F) indicated a strong Zn preference for FcaMT1, which was corroborated by the absence of glycosylated forms when produced under Zn surplus conditions due to a high structuration level of the FcaMT1 coordinating Zn^2+^ ions (Figure 3E) [27]. Based on these results, we concluded that FcaMT1 showed a clear Zn-thionein behavior.

### 2.2. Characterization of the α and δ Domains of Cephalopoda and Caudofoveata MTs

In order to characterize the biochemical properties and metal-binding features of the predicted α (from Met_1_ to Thr_44_) and δ (from Met_1_ to Lys_45_) domains of NpoMT1 and FcaMT1, respectively, we followed the same strategy as for the full-length proteins. In brief, both putative domains were expressed as GST-fusion proteins in *E. coli* BL21 cultures supplemented with ZnCl_2_, CdCl_2_ or CuSO_4_, and metal–protein complexes were purified and characterized by ICP-AES and ESI-MS analyses. The experimental masses corresponding to the apo-α and apo-δ domains (4872.0 Da and 4821.0 Da, respectively) were consistent with their theoretical masses (4872.57 Da and 4821.59 Da, respectively) (Figure 2B,D), confirming the identity of both recombinantly produced domains. 

The α domain of NpoMT1 produced in Cd-enriched culture medium yielded a unique species with four Cd^2+^ ions (Cd_4_-NpoMT1, Figure 3G), whereas when it was produced in Zn surplus conditions, it rendered a mixture of Zn_3_- and Zn_4_-α complexes (Figure 3H), suggesting a higher preference for Cd^2+^ than for Zn^2+^ ions. The presence of glycosylated MT species under Zn surplus conditions (peaks denoted by an asterisk in Figure 3H), and the yield of homometallic Cu-MT complexes when expressed in Cu-enriched medium (Figure 3I) supported the Cd-specificity of the α domain when expressed alone. In contrast, for the δ domain of FcaMT1, a unique metal-domain species (Zn_5_-δFcaMT1) was observed when it was synthesized in Zn-enriched medium (Figure 3K), and two species, a major Cd_5_-δFcaMT1 and a minor Cd_6_-δFcaMT1, when it was produced in Cd surplus conditions (Figure 3E). These results, together with the absence of glycosylated forms and with the formation of heterometallic Zn/Cu-δ domain complexes in Cu-enriched preparations (Figure 3L) sustained a Zn preference of δFcaMT1 when produced alone. In summary, our results showed that α and δ domains yield unique Cd^2+^- and Zn^2+^-complexes, respectively (Figure 3G,K), indicating that they form single and structurally well-defined metal clusters, and supporting that α and δ domains are functionally autonomous MT domains.

## 3. Discussion

### 3.1. Cephalopoda NpoMT1 and Caudofoveata FcaMT1, and the α and δ Domains

Metallothioneins are becoming a valuable model system to investigate the origin and evolution of protein domains for two main reasons. First, based on their patterns of Cys motifs, MT domains can be predicted in broad phylogenetic contexts, and thereby, the changes in their domain organization can be visualized along the evolutionary ladder. Second, standardized protocols of analytical techniques (ESI-MS and ICP-AES) allow the functional characterization of the predicted MT domains based on their metal-binding features, that is, their metal-binding preferences and capacities, exposing the functional consequences of the innovations in domain rearrangements. Several mollusk MTs and some domains had previously been functionally characterized in the Bivalvia and Gastropoda classes, but nothing was known about the MTs from the Cephalopoda class of the Conchifera clade, nor from any Caudofoveata MTs of the Aculifera clade. We have now characterized the putative MTs from Cephalopoda *N. pompilius* (NpoMT1) and Caudofoveata *F. caudatus* (FcaMT1) species, and their distinctive α and δ domains. We determined the MT features of both MTs, establishing that NpoMT1 and its α domain render homometallic complexes with seven and four Cd^2+^ ions, respectively (Figure 3A,G), whereas FcaMT1 and its δ domain coordinate eight and five zinc ions, respectively (Figure 3E,K). Considering that β1 domains bind three divalent metal ions [21,24], the metal-to-protein stoichiometries of the α and δ domains fully agree with those of the full-length MTs: 4(α) + 3(β1) = 7Cd^2+^ ions for NpoMT1, and 5(δ) + 3(β1) = 8Zn^2+^ ions for FcaMT1. These stoichiometries show a higher metal-binding capacity of FcaMT1 in comparison with NpoMT1 due to the two extra Cys at its N-terminal domain: 14 Cys in the δ domain versus 12 Cys in the α domain. Our results also indicate that the α and δ domains are functionally autonomous, and experimentally confirm the predicted bidomain αβ1 and δβ1 structures of Cephalopoda and Caudofoveata MTs, respectively.

Importantly, our results revealed a different metal preference for Cephalopoda and Caudofoveata MTs, Cd specificity for NpoMT1 and Zn specificity for FcaMT1. These metal preferences coincide with those of their N-terminal domains (Cd^2+^ ions for the α domain, and Zn^2+^ ions for the δ domain) indicating that the distinctive N-terminal domain is fundamental for the metal preference of the full-length proteins. From an evolutionary perspective, considering the ancient origin of the α domain and the pervasiveness of Cd-specific MTs in mollusks [21,30], our results suggest that the Zn-specific FcaMT1 was a functional innovation in the *F. caudatus* lineage, and thereby, likely associated to the appearance of the δ domain in the Caudofoveata class. Because marine mollusk species typically have Cd-thioneins [30], we investigated the possibility that Caudofoveata species might have an additional “conventional” αβ1- or β2β1-MT with Cd preference. Database surveys did not retrieve any putative Caudofoveata MT supporting this possibility, although we cannot definitively rule it out because the available Caudofoveata databases are still incomplete. Interestingly, however, we identified a second putative MT in *F. caudatus*, FcaMT2, made up of a “degenerated” δ-like domain (48% identical to the δ domain of FcaMT1) maintaining only 9 of the 14 Cys, fused to a conserved β1 domain (76% identical to the β1 domain of FcaMT1) (Appendix A). Whether FcaMT2 is a Zn-specific MT or it may be a Cd-thionein cannot be inferred from sequence analysis, and it requires further investigation.

### 3.2. Biological Implications of the Cephalopoda and Caudofoveata MTs

In some cases, it has been possible to relate the chemical findings on the metal-binding properties of some MTs with their biological functions. This has been achieved, for instance, in the case of the Cd- and Cu-specific MT isoforms of the terrestrial snails *Helix pomatia* and *Cornu aspersum.* The chemical properties of these MTs agree with the biological findings on their organ- and cell-specific accumulation and distribution as well as with their presumed role in the metal metabolism of these two snail species [31]. In the case of the MTs analysed in this work, although there are no data on the metal metabolism of *N. pompilius*, results from other cephalopod species consistently indicate that cephalopods preferentially accumulate Cd in their midgut gland, where they can detoxify it by binding to “metallothionein-like proteins”, including metallothioneins [29,32,33]. The closely related species *Nautilus macromphalus*, for example, has been reported to accumulate huge amounts of many different metals in its digestive gland, which may serve as a storage and detoxification organ for most of them too, probably by binding to MTs and by intracellular compartmentalization [32]. Interestingly, another study showed that some cephalopod species collected near the submarine volcano Tagoro (Canary Islands) show highly elevated concentrations for nine different metals, including Cd, apparently due to volcanic emissions of metal-containing compounds [34]. Against this background, it is therefore highly comprehensible that cephalopods have Cd-specific MTs. In contrast, biological data that could explain the analytically proven zinc preference of the metallothionein of *F. caudatus*, are completely lacking. This is mainly because there are hardly any studies on metal accumulation in Caudofoveates. One reason for this is probably that the class Caudofoveata is small and most representatives of this class live burrowing in the seabed and in deep-sea sediments [35], which makes it very difficult to perform, for example, metal exposure experiments with these animals in the laboratory. However, based on our results, it is not unreasonable to expect that zinc plays an important role in the physiology and ecology of this peculiar group of animals.

### 3.3. Origin and Evolution of MT Domains

Based on the phylogenetic distribution of the different MT domains, it is thought that the origin of α and β1 domains predated the diversification of the Mollusca phylum, while the δ domain would have evolved de novo, probably concurring with the appearance of the Caudofoveata class [21]. De novo evolution events in the MT world seem to have been more frequent than previously thought. The lineage-specific γ and δ domains in mollusks ([21] and this work), the evolution of a new 12C domain in appendicularian tunicates [36,37] or the recent evolutionary origin proposed for the yeast metallothionein CUP1 [38] sustain this idea, and support the polyphyly of MTs implicitly suggested in diverse evolutionary studies [19,20,39,40]. This polyphyly implies that although MTs form a seemingly homogenous group of proteins, they are not, and the domains or proteins grouped in the MT family would be, in fact, the result of the convergent evolution of independent sequences driven by the chemical requirements of metal coordination such as a high content of coordinating residues (i.e., cysteines) and a relatively small length that would favor its proper folding [20,21,30].

We do not know the source of the new MT domains or proteins, but in the last years data have accumulated showing that short open reading frames (sORFs) in long non-coding RNAs (lncRNAs) might be an important source of de novo peptides [41,42,43]. These sORFs have a median size of around 43 amino acids (although some of them might encode peptides longer than 100 amino acids) [44], and because they are lineage-specific, they are poorly conserved across species [41,42]. Both features, size and conservation, match those of the de novo MT domains well. The δ domain of FcaMT1 or the γ domain of LgiMT2, for instance, are 45 amino acid peptides that are specific to Caudofoveata and Patellogastropoda lineages, respectively [21,30]; the 12C domain of OdiMT1 is an appendicularian-specific peptide of 42 amino acids, different from the 12C domains of other chordate MTs [36,37]; and yeast CUP1 is a genus-specific MT [38] with its cysteines distributed throughout 44 of its 61 amino acids [45]. It is not too daring to hypothesize, therefore, that the sORFs of lncRNAs might have been an important source of novel MT domains or proteins.

### 3.4. MT Modularity and Evolvability

Protein modularity accelerates the evolutionary process of generating genetic diversity because it favors the domain shuffling, which is an effective means of exploring new viable sequences and creating new proteins as it has a greater impact on the functional properties of proteins than the progressive accumulation of point mutations in their sequences [11,46]. Modularity therefore increases protein evolvability by favouring the emergence of new protein variants that can be adaptively selected ([47] and references therein). Our analysis of NpoMT1 and FcaMT1 supports that the modular architecture of MTs contributed to their evolvability because it provides experimental evidence that the diverse mollusk MTs came from the rearrangements of structurally intact and functionally autonomous domains. Our results also show that domain rearrangements might involve not only preexisting domains (e.g., α and β1 domain) but also de novo domains (e.g., δ domain), raising the number of distinct domains available to create MTs with new functional abilities (Figure 4). 

Altogether, our results substantiate that the modular structure of MTs would increase their evolvability by facilitating the appearance of new forms (e.g., δβ1-FcaMT1) with different metal-binding properties (i.e., a higher metal-binding capacity and a new preference for Zn^2+^ ions) that would have been selected depending on the species-specific adaptive requirements in the Caudofoveata lineage (e.g., new physiological or ecological roles of zinc in the Caudofoveata lineage). Furthermore, the modular architecture of MTs would also have facilitated the duplications and losses of complete domains in a lineage-specific manner. For instance, losses of α and β domains occurred during the evolution of some mollusk lineages, including the Caudofoveata species, while these domains have been tandem duplicated in several gastropod and bivalve species, giving rise to large multidomain forms (Figure 4) [21,24,37,48,49]. In fact, duplications and losses of complete domains have also been described in MTs from other organisms, including species of the Chordata phylum [36,37,50]. In summary, it can be concluded that working as structurally and functionally independent modules, MT domains have been recurrently gained, lost, modified or recombined in mollusks and in other animal phyla, and therefore it is not unreasonable to assume that by favoring structural and functional innovation, the modular architecture of MTs has played a key role in the origin and evolution of most metazoan MTs.

## 4. Materials and Methods

### 4.1. Production and Purification of Recombinant Metal-MT Complexes

Production and purification of recombinant metal-MT complexes of NpoMT1, FcaMT1, αNpoMT1 and δFcaMT1 domains were performed as described elsewhere [37,50]. In brief, synthetic cDNAs codifying the different constructs were provided by Synbio Technologies (Monmouth Junction, NJ, USA), cloned in the pGEX-4T-1 expression vector (GE Healthcare, Chicago, IL, USA) and transformed in protease-deficient *E. coli* BL21 strain. Metal-MT complexes were produced in *E. coli* BL21 cultures expressing the recombinant plasmids, after induction with isopropyl-β-D-thiogalactopyranoside (100 μM) and supplementation with ZnCl_2_ (300 μM), CdCl_2_ (300 μM) or CuSO_4_ (500 μM). Metal-MT complexes were purified from the soluble protein fraction of sonicated bacteria by affinity purification of the GST-tagged proteins, and digestion with thrombin. The metal-MT complexes were concentrated with a 3 kDa Centripep Low Concentrator (Amicon, Merck), and fractionated on a Superdex-75 FPLC column (GE Healthcare) equilibrated with 20 mM Tris-HCl, pH 7.0 and run at 0.8 mL min^−1^. The protein-containing fractions, identified by their absorbance at 254 nm, were pooled and stored at −80 °C until use.

### 4.2. Analysis of Metal-MT Complexes

All designed constructs, NpoMT1, FcaMT1, αNpoMT1 (from Met_1_ to T_44_) and δFcaMT1 (from Met_1_ to K_45_), were characterized by means of mass spectrometry (ESI-MS) and spectroscopy (ICP-AES). An electrospray ionization mass spectrometry (ESI-MS) MicroTOF-Q Instrument (Bruker Daltonics Gmbh, Bremen, Germany) interfaced with a Series 1100 HPLC pump (Agilent Technologies, Santa Clara, CA, USA), was used to determine the molecular mass of the recombinant proteins. The instrument was calibrated with ESI-L Low Concentration Tuning Mix (Agilent Technologies) and the experimental conditions were set up as follows: injection of 10–20 μL of sample through a PEEK long tube (1–1.5 m × 0.18 mm i.d.) at 30–50 μL min^−1^; capillary-counterelectrode voltage, 3.5–5.0 kV; desolvation temperature, 90–110 °C; dry gas, 6 L min^−1^. Data were acquired over an m/z range of 800 to 3000. The liquid carriers were a 90:10 mixture of 15 mM ammonium acetate and acetonitrile at pH 7.0 and a 95:5 mixture of formic acid and acetonitrile at pH 2.4.

Element concentrations of S, Zn, Cd and Cu in the sample were determined by Inductively Coupled Plasma Atomic Emission Spectroscopy (ICP-AES) by means of a PerkinElmer Optima 4300DV (Waltham, MA, USA) at the correct wavelength (S, 182.04 nm; Zn, 213.86 nm; Cd, 228.80 nm; Cu, 324.80 nm) under conventional conditions [51]. MTs’ concentration was calculated based on the S concentration obtained by ICP-AES, assuming that all the sulfur measured comes from peptides’ Cys and Met residues.

## Figures and Tables

**Figure 1 ijms-23-15824-f001:**
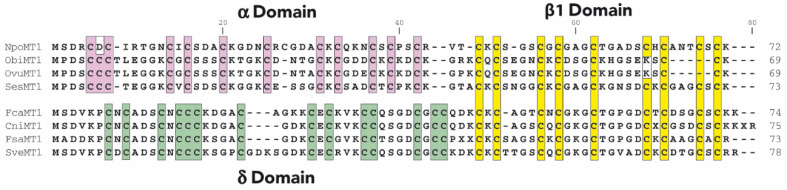
Amino acid alignments of Cephalopoda and Caudofoveata MTs. At the top of the figure, alignment of *Nautilus pompilius* (NpoMT1), *Octopus bimaculoides* (ObiMT1), *Octopus vulgaris* (OvuMT1) and *Sepia esculenta* (SesMT1) MTs showing the bidomain αβ1 structure of Cephalopoda MTs. At the bottom, alignment of *Falcidens caudatus* (FcaMT1), *Chaetoderma nitidulum* (CniMT1), *Falcidens sagittiferus* (FsaMT1) and *Scutopus ventrolineatus* (SveMT1) MTs showing the δβ1 structure of Caudofoveata MTs. Cysteines are highlighted in pink in Cephalopoda α domains, in green in Caudofoveata δ domains and in yellow in the β1 domains shared by both groups of species. Sequences are from [21].

**Figure 2 ijms-23-15824-f002:**
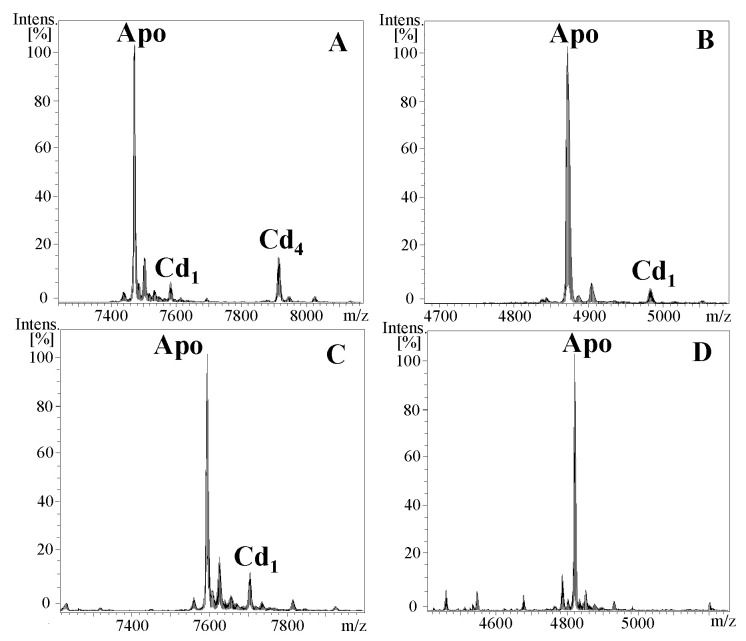
Deconvoluted ESI-MS spectra of the apo-MTs obtained after the demetallation by acidification (pH 2.4) of recombinant Cd-protein complexes. Notice that in the production process, the digestion with thrombin of the GST-MT fusion proteins resulted in the addition of two extra residues (glycine and serine) at the N-terminal end of the purified MTs that, as shown in previous studies, did not interfere with the metal-binding features of recombinant MTs [25,26]. (**A**) apo-NpoMT1, (**B**) apo-αNpoMT1, (**C**) apo-FcaMT1 and (**D**) apo-δFcaMT1.

**Figure 3 ijms-23-15824-f003:**
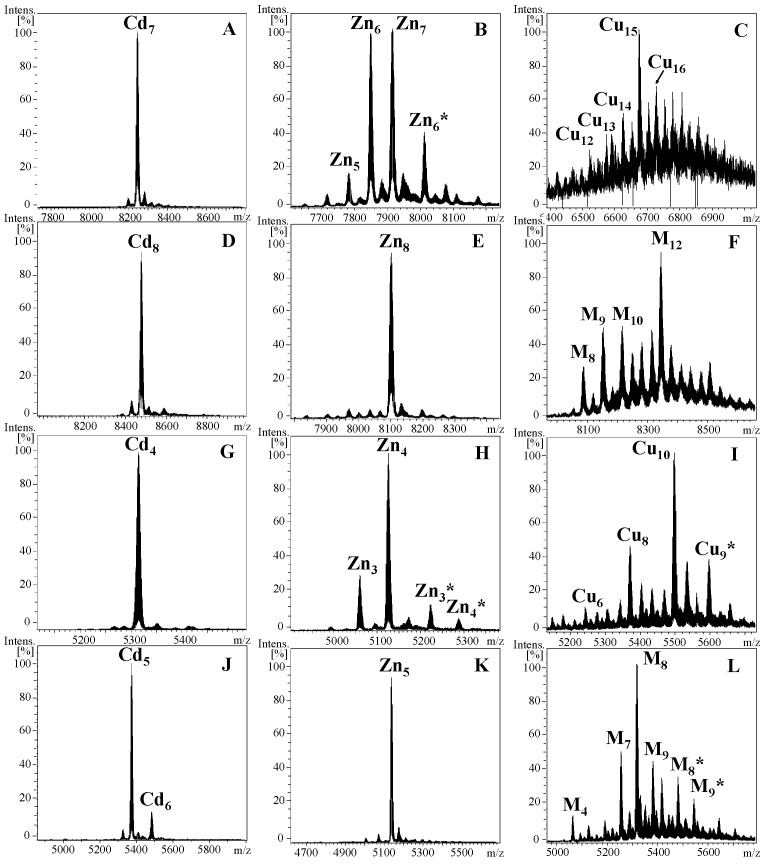
Deconvoluted ESI-MS spectra of metal–protein complexes recombinantly produced in *E. coli* growth in metal-enriched culture medium. (**A**) Cd-NpoMT1, (**B**) Zn-NpoMT1, (**C**) Cu-NpoMT1, (**D**) Cd-FcaMT1, (**E**) Zn-FcaMT1, (**F**) Cu-FcaMT1, (**G**) Cd-αNpoMT1, (**H**) Zn-αNpoMT1, (**I**) Cu-αNpoMT1, (**J**) Cd-δFcaMT1, (**K**) Zn-δFcaMT1 and (**L**) Cu-δFcaMT1. M in panels F and L stands for heterometallic (Zn + Cu)-MT complexes. Glycosylated metal-MT species are depicted with an asterisk (*).

**Figure 4 ijms-23-15824-f004:**
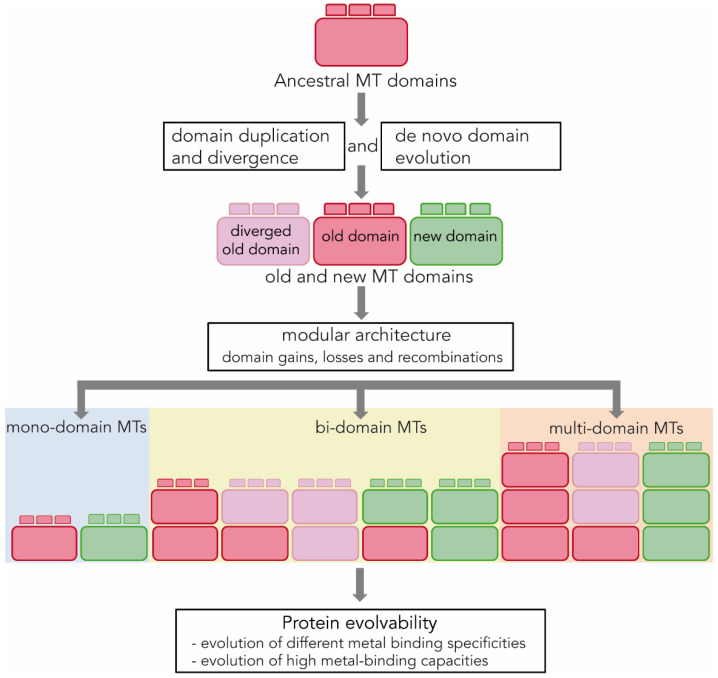
Modular architecture and MT evolvability. MTs are proteins made up of self-stabilizing, independent folding and structurally compact modules called “domains”. Some MT domains are old and date back to the metazoan origins (red), but others are more recent and would have arisen by duplication and divergence mechanisms (pink) or by de novo evolution events (green) during the diversification of different animal lineages. Most animal MTs, including those of mollusks and chordates, are bidomain proteins. This modular architecture favors the rearrangements of old and new domains leading to novel domain combinations in bidomain forms, or to domain gains and losses in multidomain and monodomain MTs, respectively. Several MTs have been described that exemplify the different rearrangements depicted in this figure [21,24,36,37,49,50]. Overall, our results suggest that the modular architecture of the MTs increases their evolvability by facilitating the appearance of new forms with different metal binding specificities (i.e., Cd-, Zn- or Cu-thioneins) or with augmented metal-binding capacities (i.e., from monodomain to bidomain and multidomain forms) that can be adaptively selected according to the metal requirements and bioavailability of the different animal lineages.

## Data Availability

Not applicable.

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
