# Peer review of "The Modular Architecture of Metallothioneins Facilitates Domain Rearrangements and Contributes to Their Evolvability in Metal-Accumulating Mollusks"

_ijms, 2022, doi:10.3390/ijms232415824_

Round 1

Reviewer 1 Report

In the manuscript “The modular architecture of Metallothioneins facilitates domain rearrangements and contributes to their evolvability in metal-accumulating mollusks”, Dr. Albalat and coworkers performed a spectrometric and spectroscopic characterization combined to phylogenetic analysis to analyze the metal-thionein nature and metal-binding preference (Cd-thionein vs Zn-thionein) of different MT domains from two uncharacterized MTs form metal accumulating mollusks. The authors then analyze their findings and compare them to known MT domains form other mollusks to draw conclusions on how the modular architecture of MTs contributed to MT evolution. The work thus provide further evidence on how domain modularity has allowed to diversify MTs and increase the ability to efficiently evolve. The study addresses an important general question in the MT field related to the importance of domains in the evolution of modular MTs with different metal preference and thionein-characters.

The manuscript is well written and easy to follow. The content and the rational approach utilized for the design of the experiments are sound, the methodology applied is nicely described, and the results properly interpreted. The findings and conclusions presented in the manuscript are comprehensive, substantiated by the results obtained, and of interest to the readership of IJMS.

I would recommend the publication of the manuscript in the current form, but I nevertheless encourage the authors to consider addressing the following point prior to publication:

-        The authors discuss the existence of glycosylation in the production of MTs in surplus zinc concentrations, based on MS data. I assume that they might have also additional characterization besides MS analysis to confirm the nature of this posttranslational modification to draw the conclusion. If it is the case I encourage the author to include this additional information in the text and material and methods.

Author Response

We thank the reviewer for his/her kind words.

Regarding the reviewer's question about glycosylation in MT production, a comprehensive analysis on MT glycosylation including Falcidens and Nautilus MT, has recently been accepted for publication in Common Chem. In this work, we demonstrate by means other than ESI-MS, such as Enzyme-Linked Lectin Assays (ELLA), that recombinant MTs are glycosylated in certain conditions. We observed 162 Da differences in MS peaks of some productions of Falcidens and Nautilus MTs that ELLA demonstrated to be the result of glycosylation of these proteins as in other cases of recombinant MT productions included in the same article (DOI: 10.1039/d2cc05589a). We have attached a copy of this article for the reviewer’s consideration.

Reviewer 2 Report

著者らは、特にアミノ末端ドメインでの頭足類とカウドフォビアタ MT の金属結合特性を分析し、明確なそれを明らかにしました。

著者らは、特にアミノ末端ドメインでの頭足類とカウドフォビアタ MT の金属結合特性を分析し、明確なそれを明らかにしました。

実験的アプローチは一般的で目新しいものではありませんが、「生きた化石」からのメタロチオネインの分析は新しいものであり、この論文でのこれらの種の使用は、MT の進化に言及したこの論文のハイライトです。そういう意味で、本誌に掲載する価値があると思います。

137 行目で、参考文献 27 は現在 SUBMITTED で、内容はわかりませんが、この原稿では非常に簡単に引用しています。27 が受理されるまで待つか、原稿にもう少し詳細を記載する方が、読者にとって親切ではないでしょうか?

図 3 (F) と (L) で、なぜ「M」の記号を金属として使用しているのですか? それはCuであるべきです。

行220~223から、本実験は、N末端ドメインが全長タンパク質の金属優先性を決定することを直接実証していない。これは、両者の C 末端ドメインの配列が必ずしも同じではないためです。N末端ドメインが全長の金属選択性を決定することがすでに一般的な知識である場合は、これを実証する論文を引用する必要があります. そうでない場合は、この研究で使用したαドメインと δドメインを交換することで、同じ実験を証明できます 。

Author Response

We thank the reviewer for finding our work interesting.

Regarding reference 27, this article has recently been accepted for publication in Common Chem. We are glad to be able to include now the full reference of this article: Garcia-Risco, M.; Gonzalez, A.; Calatayud, S.; Lopez-Jaramillo, F.; Pedrini-Martha, V.; Albalat, R.; Dallinger, R.; Dominguez-Vera, J.M.; Palacios, O.; Capdevila, M. Glycosilation in Recombinant Metallothioneins. Chem Commun (Camb) 2022, doi: 10.1039/d2cc05589a. We have attached a copy of this article for the reviewer’s consideration.

Concerning the use of the 'M' symbol for metal, instead of Cu in Figure 3 (F) and (L), the nomenclature is correct because the bound metal is not only Cu. We performed ICP-OES experiments on these samples, and they showed presence of both Zn and Cu (i.e. they are heteronuclear metal-MT complexes). Since these elements have a similar atomic weight, it is inaccurate to assign an exact number (stoichiometry) of each metal to that metal-MT complexes. For that reason, these peaks are identified as Mx where x is the number of metals that are a mixture of Cu and Zn.

Finally, the reviewer argues that “…(lines220-223), this experiment does not directly demonstrate that the N-terminal domain determines the metal preference of the full-length protein. This is because the sequences of her C-terminal domains in both are not necessarily the same. If it is already common knowledge that the N-terminal domain determines full-length metal selectivity, the paper demonstrating this should be cited”. To our knowledge, there are no experiments showing that N-terminal domains determine full-length metal selectivity, and it was not our idea to suggest this possibility. What is said in the sentence of the article is “…our results have revealed a different metal preference for Cephalopoda and Caudofoveata MTs, Cd-specificity for NpoMT1 and Zn-specificity for FcaMT1. These metal preferences coincide with that of their N-terminal domains (Cd2+ ions for the α domain, and Zn2+ ions for the δ domain) indicating that the distinctive N-terminal domain is fundamental for the metal preference of the full-length proteins”. We think that based on the coincidence in the metal preference of N domains and full-length proteins, it is safe to assume that the N-terminal domain contributes to the metal preference of full-length proteins, which does not mean that N domains uniquely determine metal preference, and of course does not exclude a contribution of the C domains in this role.

Reviewer 3 Report

The manuscript by Calatayud et al shows the metal preference of two molluscs metallothioneins (MT) using analytical spectrometric and spectroscopic techniques. Authors demonstrate the preference for cadmium of Nautilus pompilius NpoMT1 and for zinc of Falcidens caudatus FcaMT1.

The research is appropriately designed and the manuscript well written in a logical order. I have just a few comments/suggestions that may help to strength authors’ claims:

• The title reflects more an speculative idea than the work described in the manuscript. It could be misleading to many readers as there is no evolutionary study or demonstration that modularity is indeed contributing to evolvability.

• Figure 3, have the authors considered expressing the MTs in a mixture of Cd and Zn to demonstrate the metal preference in competing conditions?

• Figure 3, panel C: how do the authors interpret the inter-peaks values? For example between Cu15 and Cu16?

• Page 5, line160: how are the hetero-metallic complex (Zn, Cu) identified? This is not clear from the text. Are the authors also including Zn in the Cu-enriched medium? Or is Zn always present?

• Page 9, line349: how do the concentrations of metals used in this study compared to what the molluscs will face in nature?

Author Response

Regarding reviewer 3's comments/suggestions:

1. The title reflects more an speculative idea than the work described in the manuscript. It could be misleading to many readers as there is no evolutionary study or demonstration that modularity is indeed contributing to evolvability.

We regret to disagree with the reviewer regarding the title. We feel that the title reflects the main message of the article because our results have demonstrated the modular architecture of MTs, i.e. that MTs are organized in independent structural and functional domains that can rearrange to create new proteins. Because MT domains might arise and recombine into new MTs that have been selected in different mollusk lineages, we think that our analyses of NpoMT1 and FcaMT1 support that the modular architecture of MTs has contributed to their evolvability (see 3.4. MT Modularity and Evolvability in the Discussion section). We would like therefore to keep the title as it is.

2. Figure 3, have the authors considered expressing the MTs in a mixture of Cd and Zn to demonstrate the metal preference in competing conditions?

Our article focuses on evolutionary and genetic aspects of MTs, such as their modular organization and the origin and evolution of the different domains. We think that the experiments proposed by the reviewer, although they are undoubtedly a good idea, they would be beyond scope of this paper and would be more appropriate for a future and more specific work focused on the chemical properties of the MTs.

3. Figure 3, panel C: how do the authors interpret the inter-peaks values? For example between Cu15 and Cu16?

These peaks correspond to dimeric species of metal-MT complexes. As an example, a dimeric form associated to 31 Cu(I) ions would display a peak between Cu15 and Cu16 peaks of the monomeric form.

4. Page 5, line160: how are the hetero-metallic complex (Zn, Cu) identified? This is not clear from the text. Are the authors also including Zn in the Cu-enriched medium? Or is Zn always present?

We identify the hetero-metallic complexes by ICP-OES, measuring the levels of the three studied metals (Cu, Zn and Cd) in all samples. In Cu-enriched medium no Zn is added, but basal levels of Zn are always present in the cell, as it is an essential element for living (doi: 10.1126/science.1060331). Thus, recombinant MTs produced in Cu-enriched medium might sequester Zn ions from the intracellular medium if Cu-MT complexes need them to stabilize.

5. Page 9, line349: how do the concentrations of metals used in this study compared to what the molluscs will face in nature?

Metal concentrations in nature under normal conditions are lower than those used in the study during the formation of metal-MT complexes. However, in exceptional situations, whether due to human activities or natural disasters, it has been shown that local concentrations of metals in soil, water, air and plants can increase significantly, and directly affect the health of organisms, or indirectly to those who consume them through the food chain. In these situations, metals accumulate in different organs and are detoxified by binding to metallothioneins. We have discussed an example of an abnormal metal concentration caused by the eruption of the submarine volcano Tagoro in the Canary Islands (see 3.2. Biological implications of the Cephalopoda and Caudofoveata MTs in the Discussion section).